# Developing Device of Death Operation (DODO) to Detect Apoptosis in 2D and 3D Cultures

**DOI:** 10.3390/cells13141224

**Published:** 2024-07-20

**Authors:** Ziheng Zhang, Zhe Sun, Ji-Long Liu

**Affiliations:** 1School of Life Science and Technology, ShanghaiTech University, Shanghai 201210, China; 2Department of Physiology, Anatomy and Genetics, University of Oxford, Oxford OX1 3PT, UK

**Keywords:** DODO, 2D culture, 3D culture, degron, apoptosis

## Abstract

The real-time detection of intracellular biological processes by encoded sensors has broad application prospects. Here, we developed a degron-based modular reporting system, the Device of Death Operation (DODO), that can monitor various biological processes. The DODO system consists of a “reporter”, an “inductor”, and a “degron”. After zymogen activation and cleavage, the degron will be released from the “reporter”, which eventually leads to the stabilization of the “reporter”, and can be detected. By replacing different “inductors” and “reporters”, a series of biological processes can be reported through various signals. The system can effectively report the existence of TEV protease. To prove this concept, we successfully applied the DODO system to report apoptosis in 2D and 3D cultures. In addition, the reporter based on degron will help to design protease reporters other than caspase.

## 1. Introduction

The techniques of detecting intracellular signals play important roles in biological research, especially for key cellular molecules representing basic biological processes. Informative detections include the detection of the expression of some key genes, the translocation of some proteins, and the activation of zymogens.

To gain insight into the physiological state of cells, the common practice is to determine the expression, location, and modification of key biomarkers through a series of biochemical experiments. In recent decades, people have developed a series of detection methods for different physiological states and signal molecules. In particular, biological coding reporting systems have been used, such as coupling conformationally sensitive cpEGFP (circularly permutated EGFP) with human dopamine receptors to detect dopamine [1]. Similar methods include calcium sensors and acetylcholine sensors [2,3]. In general, these methods are tailored to the characteristics of different targets.

Signals of many intracellular events are transmitted through the activation of zymogens. A classic case based on zymogen activation is apoptosis, which is performed by caspase [4,5]. Apoptosis has always been the focus of attention because it is related to developmental events, cancer treatment, and many diseases [5,6]. In recent years, people have developed a series of methods to detect apoptosis, mainly by detecting the activation of key signaling molecules, such as caspase-3. The activation of zymogens plays a fundamental role in many intracellular pathways, which means effectively monitoring the activation of zymogens is particularly important for understanding a range of biological processes.

Some natural amino acid sequences that have been found to regulate the degradation of target proteins are called degrons [7]. It has been reported recently that a series of degrons is mainly used to artificially regulate protein degradation [8,9,10]. A famous example is AID (auxin-inducible degron), a powerful tool developed from auxin receptors in plants [8]. F-box transport inhibitor reaction 1 (TIR1) is a receptor for auxin. By binding to auxin, chimeric E3 ligase (SCF-TIR1) recruits and ubiquitinates the degron [11,12]. This is a natural and rapid method of regulating substrate proteins. Fortunately, the orthologs of TIR1 are only found in plants, which provides the possibility to apply it to other species. Based on this principle, AID has developed into a powerful technology for the conditional and rapid degradation of target proteins. By exogenously expressing TIR1 and adding auxin or other analogs to the medium, such as IAA (indole-3-acetic acid) and NAA (1-naphthylacetic acid), the degron-labeled protein will degrade rapidly [8,13]. Driven by gene knock-in technology based on CRISPR/Cas9, it is widely used for rapid degradation of endogenous proteins [13,14].

In addition, many proteins have naturally unstable motifs at the N-terminal or C-terminal, which may be composed of a few amino acids. Large-scale N-terminal screening or C-terminal screening can be used to characterize these unstable motifs [15,16]. These results have been confirmed by some natural proteins and can be used to develop a series of tools [17,18]. This is a suitable element for muting the reporting signal in the control unit. In this report, we introduce the concept of modularity, which combines a “reporter”, an “inducer”, and a “degron” into a complete reporting system, which we name “the Device of Death Operation (DODO)”. DODO can be used to report the activity of target enzymes in living cells and be applied for identifying apoptosis in vivo. Moreover, people can change these elements according to the needs of different purposes.

## 2. Materials and Methods

### 2.1. Cell Culture

For HEK293T, cells were grown in DMEM (Hyclone, cat. SH30022.01, Cytiva, Wilmington, DE, USA) containing 10% FBS (Biological Industries, cat. 04-001-1A, Kibbutz Beit-Haemek, Israel) in a humidified incubator at 37 °C with 5% CO_2_. For MCF-10A, cells were grown in Growth Medium, consisting of DMEM/F12 (Corning, cat. 10-092-cvr, Corning, NY, USA)) with 10% House Serum (Biological Industries, cat. 04-124-1A), 20 ng/mL EGF (Peprotech^®^, cat. AF-100-15-100, Thermo Fisher, Waltham, MA, USA), 0.5 mg/mL Hydrocortisone (Selleckchem, cat. S1696, Houston, TX, USA), 100 ng/mL Cholera Toxin (APExBIO, cat. B8326, Houston, TX, USA), and 10 μg/mL Insulin (MedChemExpress, cat. HY-P1156, Monmouth Junction, NJ, USA), in a humidified incubator at 37 °C with 5% CO_2_.

### 2.2. Plasmid Construction

All plasmids were created by standard molecular biology techniques and confirmed by exhaustively sequencing the cloned fragments. In particular, for pLV-ZsGreen1-IRES-mCherry-TEV_cleavage_site-RRRG, we introduced the TEV cleavage site (5′-GAGAACCTGTACTTCCAGAGC-3′, amino acid sequence ENLYFOS) between the C-terminal of mCherry and the degron using a ClonExpress Ultra One Step Cloning Kit (Vazyme Biotech, cat. C115-01, Nanjing, China). The same approach was used to construct pLV-ZsGreen1-IRES-mCherry-TEV_cleavage_site (without the RRRG degron). For the plasmids to report active caspase-3, the pLV-ZsGreen1-IRES-mCherry-Caspase3_cleavage_site-RRRG was constructed based on pLV-ZsGreen1-IRES-mCherry-TEV_cleavage_site-RRRG, replacing the TEV cleavage site with the caspase-3 cleavage site (5′-GGAGACGAGGTGGACGGC-3′, amino acid sequence GDEVDG). Another vector used for reporting activated caspase-3 was pLV-mCherry-caspase3_cleavage_site-RRRG-IRES-EYFP, implemented by inserting the synthetic *mCherry-caspase3_cleavage_site-RRRG* sequence before IRES (Appendix A).

### 2.3. Lentivirus Packaging and Stable Cell Line Construction

HEK293T cells used for lentivirus packaging and all plasmids used for lentivirus packaging were prepared using an E.Z.N.A.^®^ Endo-free Plasmid DNA Mini Kit (Omega Bio-tek, cat. D6950-01B, Norcross, GA, USA) and transfected with Lipofectamine 2000 (Thermo Fisher, cat. 11668019, Waltham, MA, USA). HEK293T cells were cultured in a 6 cm dish and co-transfected with packaging vector psPAX2 (Addgene#12260), envelope vector pMD2.G (Addgene#12259), and transfer vector pLV-ZsGreen1-IRES-mCherry-TEV_cleavage_site-RRRG, pLV-ZsGreen1-IRES-mCherry-caspase3_cleavage_site-RRRG, or pLV-mCherry-caspase3_cleavage_site-RRRG-IRES-EYFP into HEK293T cells in a ratio of 3:1:4 (totaling 5 μg for a 6 cm dish) when the cell density was about 70%. Forty-eight hours after transfection, the viral supernatant was collected and filtered through a 0.45 μm PES filter. A total of 1 mL of virus supernatant was used for cell infection. Seventy-two hours after infection with lentivirus, the cells that were EGFP-positive and mCherry-negative were collected by FACS.

### 2.4. Immunofluorescence

For the immunofluorescence experiment, the HEK293T-ZsGreen1-IRES-mCherry-TEV_cleavage_site-RRRG cells were transfected with pcDNA3.1-TEV protease-myc. Twenty-four hours after transfection, the cells were washed with PBS and fixed through 4% PFA at room temperature for 15 min. A total of 5% BSA was used for blocking and the cells were incubated with anti-myc (Santa Cruz Biotechnology, cat. sc-40, Dallas, TX, USA, 1:200 dilution) antibody overnight at 4 °C. After being washed with PBST (PBS with 0.1% TritonX-100, Sigma-Aldrich, cat. T8787, St Louis, MO, USA), the cells were incubated with the secondary antibody (Jackson ImmunoResearch, cat. 715-175-151, West Grove, PA, USA, 1:500 dilution) at room temperature for 1 h, after mounting the slides used for imaging.

### 2.5. Target Signal Response Assay

For the response assay of TEV protease, the HEK293T-ZsGreen1-IRES-mCherry-TEV_cleavage_site-RRRG cells were transfected with pcDNA3.1-TEV protease–myc. Twenty-four hours after transfection, the cells were used for fluorescence measurements. For FACS measurements, the cells were washed with PBS and then collected by centrifugation after trypsinization. The collected cells were passed through a 45 μm cell sieve before measurement by FACS (BD Biosciences, FACSAria™ III, Franklin Lakes, NJ, USA) with a 561 nm laser, and the collected data were analyzed by FlowJo. For imaging measurements, the cells were fixed by 4% PFA at room temperature for 15 min after washing by PBS and then imaged after mounting. The fluorescence intensity was analyzed by Fiji.

### 2.6. 3-Dimensional Culture of MCF-10A

Before 3D culture, MCF-10A was grown in Growth Medium, consisting of DMEM/F12 (Corning,10-092-cvr) with 10% House Serum (Biological Industries, cat. 04-124-1A), 20 ng/mL EGF (Peprotech, cat. AF-100-15-100), 0.5 mg/mL Hydrocortisone (Selleckchem, cat. S1696), 100 ng/mL Cholera Toxin (APExBIO, cat. B8326), and 10 μg/mL Insulin (Medchemexpress, cat. HY-P1156) in a humidified incubator at 37 °C with 5% CO_2_. For 3D culture, the cells were digested by 0.05% Trypsin-EDTA and resuspended in 3–5 mL Resuspension Medium (DMEM/F12 with 10% House Serum). The mixture was centrifuged to harvest the cells and resuspend in 1 mL of Assay Medium (DMEM/F12 with 10% House Serum, 0.5 mg/mL Hydrocortisone, 100 ng/mL Cholera Toxin and 10 μg/mL Insulin). The, the cells were filtered with a cell strainer and counted to achieve a cell concentration of 12,500 cells/mL in Assay Medium with 2% Matrigel (Corning, cat. 354230) and 5 ng/mL EGF. We plated 400 μL of this mixture on top of the solidified Matrigel in each well of the 8-well chamber slide (Cellvis, cat. C8-1.5H-N, Mountain View, CA, USA). This corresponded to a final concentration of 5000 cells/well in an Assay Medium containing 2% Matrigel and 5 ng/mL EGF. We allowed the cells to grow in an incubator at 37 °C with 5% CO_2_. The cells were refed with Assay Medium containing 2% Matrigel and 5 ng/mL EGF every two days. The cells were used for imaging at the desired time point.

## 3. Results

### 3.1. Strategies for Developing DODO

As a modular reporting system, we designed DODO into three components according to their functions, a “reporter”, an “inductor”, and a “degron”. Among them, the “reporter” is used to report signals, the “inductor” is used to monitor specific signals, and the “degron” is used to trigger its degradation. When there is no target signal in the cell, it will trigger the degradation of “reporter” and make the cell negative. Otherwise, the “inductor” will cut off the degron, stabilize the “reporter”, accumulate in the cells, and, finally, be detected (Figure 1A). In this system, the “reporter” can be determined according to the user’s needs, such as fluorescent protein, luciferase, or other labels. The “inductor” depends on the target signal, for example, the apoptosis inductor we used in this study was the caspase-3 cleavage sequence (GDEVDG). For the degron, we used the RRRG degron in C-terminal, which, despite having only four amino acids, was shown to be sufficient to direct the tagged proteins to proteasomes for degradation [17].

Background noise is usually an important factor that makes it impossible to accurately determine the authenticity of the target signal. Therefore, we first determined the degradation ability of the RRRG degron. We used fluorescent protein to achieve this by adding RRRG degron at the C-terminal of a fluorescent protein to trigger the degradation of the fluorescent protein, thereby determining its degradation ability. In particular, the degradation ability of the RRRG degron was determined by comparing the fluorescence intensity of mCherry with and without the RRRG degron in the C-terminal. For quantification, a green fluorescent protein (ZsGreen1) was used as a control and linked to mCherry, driven by the same promoter via an IRES (internal ribosome entry site) (Figure 1B,C). By comparing the mCherry fluorescence intensity with or without the RRRG degron, it can be determined that the RRRG degron can degrade about 95% of the target protein (Figure 1D). This demonstrates the powerful ability of the RRRG degron to degrade target proteins and further indicates that it is suitable for the development of this modular reporting system.

### 3.2. DODO Responds to Target Signals

To test whether the DODO system can respond to specific signals, we introduced a TEV protease recognition and cleavage sequence between the “reporter” and “degron” as an “inductor” and detected whether the presence of TEV protease can be reported. TEV protease has categoric sites and high specificity and has been widely used in a series of applications including protein purification. The presence of TEV protease was detected and reported by introducing the recognition and cleavage sequence of TEV protease between the “reporter” and “degron” (Figure 2A).

To ensure the consistency of data, we constructed a monoclonal cell line with stably expressed ZsGreen1-IRES-mCherry-TEV_cleavage_site-RRRG to verify whether the system can respond to TEV protease. By transiently transfecting pCMV-TEV protease-myc into a stable cell line, the successfully transfected cells showed mCherry-positive results (Figure 2B). Further flow cytometric detection of cells after pCMV-TEV protease-myc transduction showed that the cells could respond to TEV protease and were mCherry-positive (Figure 2C).

### 3.3. DODO Detects Apoptosis in 2D Culture

The above experiments have confirmed that DODO can effectively respond to the emergence of target signals. We further applied DODO to the report of specific signals in cells. There are many kinds of signal molecules with protease activity in cells, among which the caspase family is the most typical and well known. We attempted to apply this modular self-degradation reporting system to report the activation of caspase-3 and ultimately whether the cells initiated apoptosis by introducing the cleavage site of caspase-3 between mCherry and the RRRG degron (Figure 3A).

To determine whether the reporting system could respond to activated caspase-3, we first used transient transfection of pseudo-activated procaspase-3 to simulate activated caspase-3 [19]. After expressing pseudo-activated procaspase-3, the cells were switched to mCherry-positive, and the intensity of the mCherry signal increased about 20 times (Figure 3B,C). Apparently, the reporter system is very sensitive to the activated caspase-3; we named this reporter system for apoptosis the Device of Death Operation (DODO).

### 3.4. DODO Detects Apoptosis in 3D Culture

After confirming that DODO can respond to activated caspase-3 in 2D cultured cells, we tried to apply it to report apoptosis under physiological conditions. As an epithelial cell line derived from the mammary gland, the MCF-10A cell line is usually used to simulate mammary-gland-like models in vitro through 3D culture. Under this condition, a single MCF-10A cell will gradually grow from a single cell to a cell cluster and finally form organoids displaying a hollow lumen. Over time, the cells located in the cell cluster will undergo apoptosis and finally form a cavity (Figure 4A).

This process is mainly divided into two stages. One is the proliferation stage, which starts from about 1~8 days. At this stage, cells mainly divide and proliferate to form cell clusters. The second stage is apoptosis (about 7~13 days). Due to the limitation of malnutrition, the cells located in the cell cluster begin apoptosis. The 3D culture of MCF-10A is a good model for reporting apoptosis [20]. We use this model for testing the sensitivity of DODO.

We constructed mCherry-caspase3_cleavage_site-RRRG-IRES-EYFP stably expressing MCF-10A cells by lentivirus infection. Matrigel was used to support a 3D culture of MCF-10A cells. We tried to obtain images of different stages, including the proliferation stage, to test whether the apoptosis reporting system would incorrectly report cells in the proliferation stage and whether it could report apoptotic cells in the apoptosis stage.

We took images of MCF-10A cell clusters on day 6 of the 3D culture (Figure 4B). Regardless of the cells located in the outer layer or the cells located in the inner layer, the cell clusters did not show a mCherry signal. This shows that the degron can maintain the ability to degrade the target protein in this environment, and it also shows that the reporting system will not misreport in the stage of cell proliferation.

We also imaged MCF-10A cells in the apoptotic state (Figure 4C). A specific internal cell showed mCherry-positive results, indicating that caspase-3 of these cells was activated. The above data show that the reporting system can effectively report the apoptosis of MCF-10A cells in 3D culture.

## 4. Discussion

Codable tools for reporting cellular biological processes are of great significance for people to identify the physiological state of cells. In particular, modular tools can provide researchers with a variety of options and reduce application restrictions.

In this report, we described a new degron-based modular reporting system, DODO, for the effective detection of a series of biological processes. DODO is mainly used for a series of intracellular signal proteins with protease activity. By introducing the target sequence of a specific signal protein with protease activity between a reporter and a degron, a report of specific intracellular signals and cell physiological state can be realized.

However, an important limitation of this tool is that it is based on the accumulation of reporters. Since the reporter signal comes from the accumulation of reporter molecules, when the target signal appears, the “reporter” can exist stably and start to accumulate. It takes a certain amount of time from the start of accumulation to reaching sufficient intensity to be detected, which may not enable reporting on the occurrence of transient signals to be conducted promptly. To make the signal of the “reporter” accumulate as quickly as possible after responding to the target signal, we usually use extremely strong promoters to drive DODO and select the highest expression cells, which introduce pressure in the cells of protein synthesis and degradation.

A previous report described a GFP-based fluorescent protease reporter gene called FlipGFP, which reverses a beta strand of the GFP. After protease activation and cleavage, beta-strand recovery leads to the reconstruction of GFP and fluorescence [21]. Recently, FlipGFP has been used as a tool for high-throughput identification of inhibitors of 3CL^pro^ of SARS-CoV-2 [22]. This provides a new strategy for reporting target signals, but there are still some limitations. Just like FlipGFP, the reporter is fixed, which leads to the unity of its detection form (although there are also red fluorescent versions). People cannot change the reporter according to their own needs. DODO, as a modular reporting system, provides researchers with wider applicability as it can be assembled into appropriate tools, according to their needs, and replaced simply.

## 5. Conclusions

We develop DODO as a codable and degron-based modular protease reporting system in this study. By changing the components of the system, different protease signals can be reported in a variety of reporting ways.

## Figures and Tables

**Figure 1 cells-13-01224-f001:**
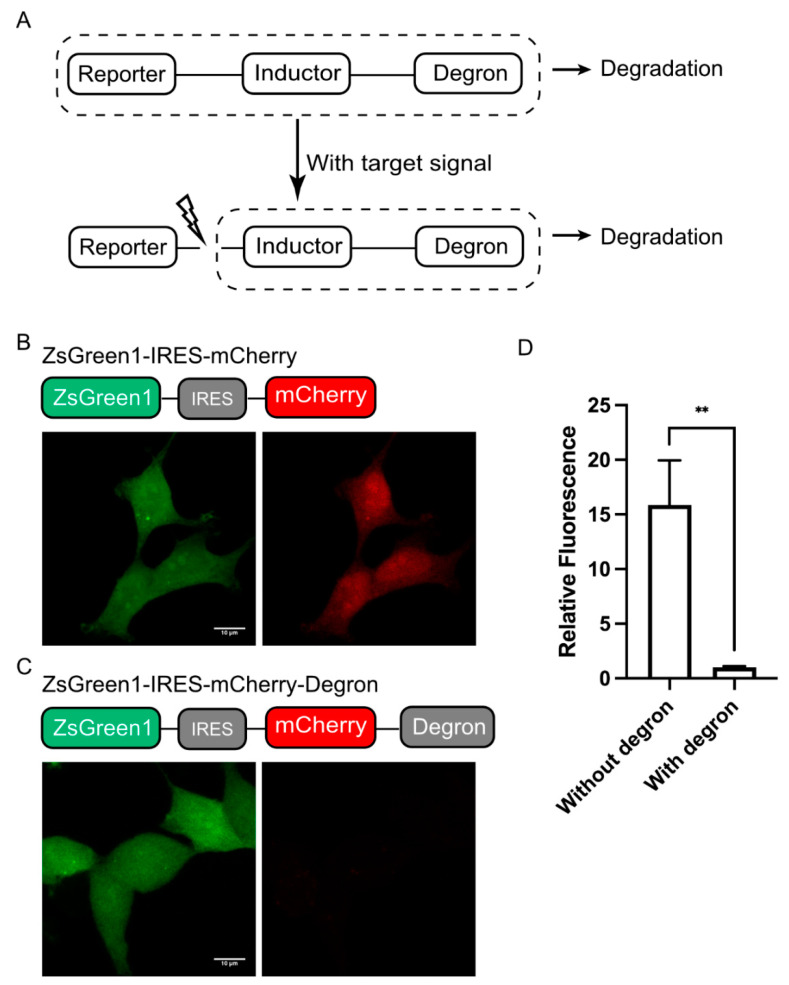
Design of DODO. (**A**) Design and working principle of the self-degradation reporting system. (**B**) The stable expression of ZsGreen1 in cells and the co-expression of mCherry mediated by IRES under the same promoter. (**C**) ZsGreen1 and RRRG-degron-labeled mCherry were stably expressed under the same promoter. (**D**) The relative fluorescence intensity of mCherry with and without the RRRG degron (*n* = 3, mean ± S.E.M. ** *p* < 0.01).

**Figure 2 cells-13-01224-f002:**
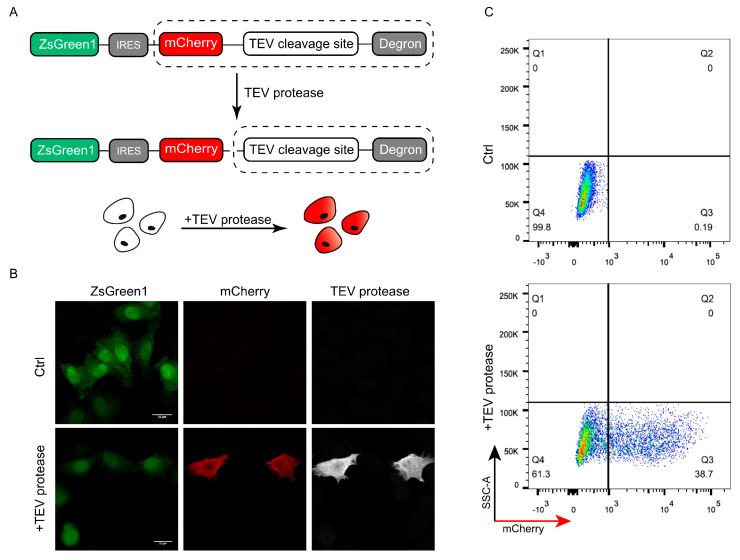
Validation of DODO by TEV protease and TEV cleavage sequence. (**A**) The pattern diagram shows the introduction of the TEV cleavage sequence between mCherry and the degron to detect whether DODO can report the presence of a TEV signal. (**B**) Fluorescence of cells stably expressing ZsGreen1-IRES-mCherry-TEV_cleavage_site-RRRG with and without TEV protease plasmid transfection. (**C**) Thirty-six hours after transfection of pCMV-TEV protease-myc, the fluorescence signal of the cells was analyzed by FACS.

**Figure 3 cells-13-01224-f003:**
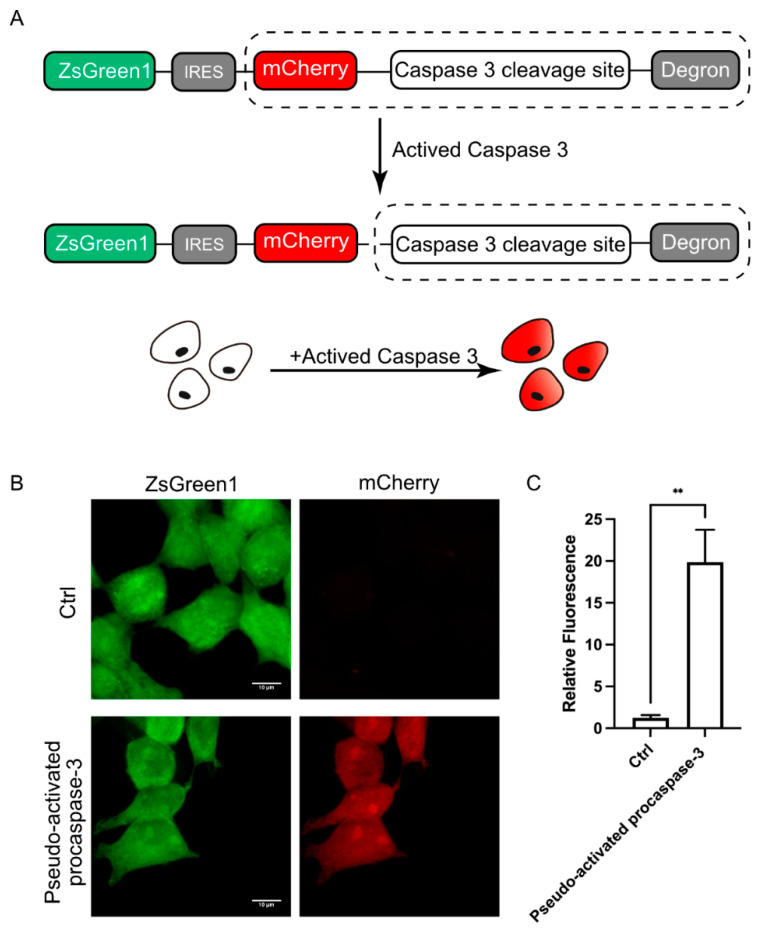
DODO responds to activated caspase-3. (**A**) The pattern diagram shows the introduction of the caspse-3 cleavage sequence between mCherry and the degron to report the activation of caspase-3. (**B**) Fluorescence of cells after transfection of pseudo-activated procaspase-3. (**C**) Relative fluorescence intensity of cells with transfected or without transfected pseudo-activated procaspase-3 plasmid (*n* = 3, mean ± S.E.M. ** *p* < 0.01).

**Figure 4 cells-13-01224-f004:**
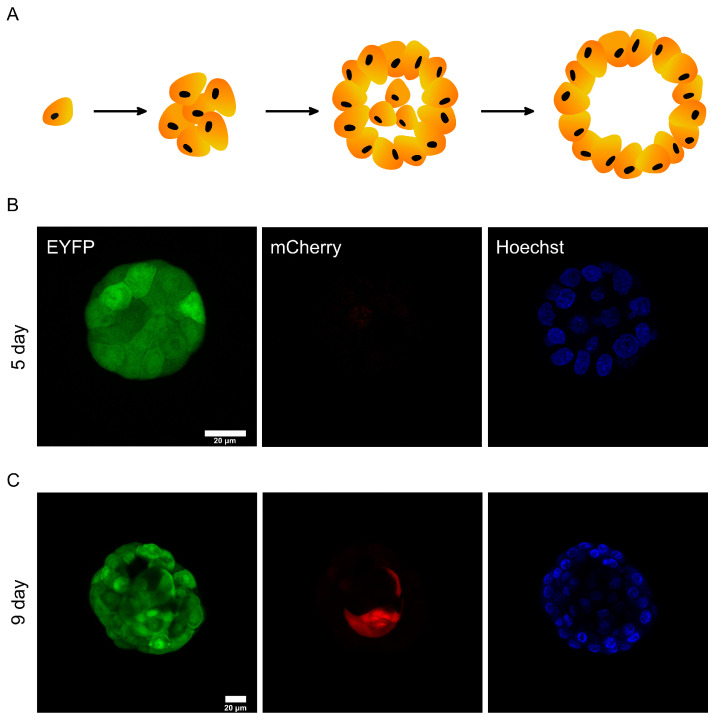
DODO determines the apoptosis of MCF-10A during metamorphic development in 3D culture. (**A**) In the schematic diagram of the metamorphic development process of MCF-10A in 3D culture, the cells in the center of the cell cluster will gradually undergo apoptosis and eventually form a cavity. (**B**) The cell cluster was proliferative without apoptosis, and all cells were mCherry-negative. (**C**) The cell cluster was in a state of apoptosis; the central cells began apoptosis and were marked by DODO as mCherry-positive.

## Data Availability

The original contributions presented in the study are included in the article/Appendix A. Further inquiries can be directed to the corresponding author.

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
