# Peer review of "Developing Device of Death Operation (DODO) to Detect Apoptosis in 2D and 3D Cultures"

_cells, 2024, doi:10.3390/cells13141224_

Round 1

Reviewer 1 Report

Comments and Suggestions for Authors

Dear authors,

greetings!

The manuscript “Developing DODO to detect apoptosis in 2D and 3D cultures” addresses the development of a degron-based modular reporting system, referred to as Device of Death Operation (DODO), to detect apoptosis.

I believe some alterations need to be performed in the manuscript before publishing it in “Cells”.

It is important that authors perform a review of the manuscript to make sure that all abbreviations, when mentioned for the first time, are followed by their meaning. TEV, cpEGFP, AID, IRES, EYFP are missing the explanation regarding meaning when presented. Italics also need to receive attention in expressions such as in vivo (for example in line 97) and in vitro (for example in line 182). When abbreviating “milliliters”, the correct term is “mL”; the L is a capital one; it is necessary to adjust that once there are places, for example in line 285, in which it is presented as “ml”.

When it comes to “Introduction”, I believe that adding the example of apoptosis in line 43 would favor comprehension. The information presented between lines 65 and 67 is confusing and needs to be rewritten to facilitate understanding. It is also important to present the definition of degron, such as the one present here: https://www.ncbi.nlm.nih.gov/pmc/articles/PMC9535385/#:~:text=Degron%20is%20a%20specific%20amino,protein%20during%20or%20after%20translation.

Regarding “Results”, the text from lines 104 to 117 should be moved to “Introduction”, as it explains the system and does not contain results. The fragment from lines 137 to 144 is confusing and should be rewritten. In lines 146 and 147 the word “stable” is present. It is a good idea to substitute one of them for a synonym or remove it. The caption of Figure 2 mentions a Figure 2D: “Relative fluorescence intensity of the cells with or without TEV”. However the image 2D is missing; it is necessary to add it. HEK293T cells are mentioned in “Materials and Methods” but not in any other section. It is necessary to present a clear state in “Results” highlighting which ones were obtained using these cells. There is no evidence to confirm the plasmid sequence in the manuscript; I suggest that authors protect the sequence (for example through a patent application) and release the sequencing results as a supplementary file. Alternatively authors may present electrophoresis gels’ images obtained during the steps of plasmid construction proving the increase in pb counting due to each sequence addition to the construction. 

“Discussion” lacks future perspectives and needs to address the large potential this device offers to different fields of expertise. Authors need to explore it in this manuscript.

When it comes to “Materials and Methods” the section lacks the description of replicates associated with each experiment; it is important to prove to readers that the data are reliable and that the study is reproducible. And is also necessary to evaluate data obtained in these replication through statistical analysis; authors need to create a section to describe the statistical test and software used and to display in figures the result of this analysis indicating statistical significant results with asterisk(s), for example. The experimental design also needs to include experiments dedicated to confirming apoptosis through another method such as flow cytometry after treatment with propidium iodide in conjunction with Annexin V; it is important to confirm DODO’s capacity to report this biological process.

Comments on the Quality of English Language

English needs to be improved. I am not a native but I could detect problems associated to conjugation (for example: the orthologs of TIR1 is only found… line 75), choice of verbal tenses (in results I could detect simultaneously past, present and future tenses), and informality (I believe it is a good idea to substitute “We” for passive voice to increase formality).

Reviewer 2 Report

Comments and Suggestions for Authors

Zhang et al. developed a degron based modular reporting system, the Device of Death Operation (DODO), which can be used to detect a series of biological processes.  They successfully applied the DODO system to report apoptosis.

The system is innovative and will get the attention of wide readers.  However, there are a couple of concerns that need to be addressed before acceptance.

1) In Fig. 4, the authors need to perform a control experiment with caspase inhibitors, such as zVAD-fmk, to prove whether the system specifically detects apoptosis.

2) A couple of detailed experimental procedures are missing. 

How was C-degron (RRRG) included in the plasmid construct? 

How did the authors perform “transient transfection of active caspase 3” (Line 167-168)? 

How did they construct “a stable monoclonal stable cell line” (Line 146-147)?

Reviewer 3 Report

Comments and Suggestions for Authors

The manuscript is focused on the detection of apoptosis in 2D and 3D models using degrons, and applying a DODO system (as the authors name it). 

The authors developed the DODO to be detected when the cell death is initiated via the caspases. 

Comments: 

- What type of plates did you used for the 3D culture? I understand that you used chamber slides, which type, which company? Please mention. 

- Why don't you tried the low binding round shaped plates for the MCF10A 3d cultures? 

- Why did you choosed 5000 cells/well? Why not less or more? Is there another protocol that you have tested, and published, or cited? 

- Somehow, the paper is not easy to follow, the subject is of interest, but I recommend some changes in the structure of the paper. In the current form I find it hard to follow, always need to move from figures, to results and back, then to the methods, and back to the content.  

I would like to see it again after structure changes in a better format. 

Reviewer 4 Report

Comments and Suggestions for Authors

The manuscript entitled “Developing DODO to detect apoptosis in 2D and 3D cultures” describes the development of a protease-mediated activation reporter based degron system to detect the apoptosis process. The manuscript can be accepted after addressing the below comments

comments :

1.      Please mention the complete form at the first occurrence Ex : line 116, 126

2.      Requires western blot confirmation for the expression reporter-based degron at different time points

3.      Real-time analysis of reporter degron after TEV/caspsase3 activation is required to verify the efficiency of DODO ( example : t0 to 24 hr)

4.      Design is limited only to the detection of apoptosis in cells, does it have a broad application? How sensitive is the reported system?

5.      How the cryptic degron is activated, Does It require activation please describe its mechanism.

6.      What happens to the cells with overexpression of caspase3, does it leads to programmed cell death?

7.      In the experimental section please mention the antibody dilutions used in the study.

8.      A quantification to compare no.of cells positive for green positive vs mcherry positive after transfection with TEV/ caspase3  would be helpful.  Are all the cells positive for mCherry? Or only a fraction of cells are positive for mCherry. Why? check Figure 2b.

9.      Figure 2b, why the Zs green has less intensity compared to ctrl? Also why the mcherry signal showed cytosolic localization? Whereas Zsgreen has cytosolic as well as nuclear localization. Add DAPI staining for Figure 2b

10.   What happens to the DODO system in the presence of drugs which induce apoptosis? It's required to extend the assay in the presence of small molecules to confirm its application in the high throughput screening of compounds.

Comments on the Quality of English Language

Minor editing of english

Round 2

Reviewer 1 Report

Comments and Suggestions for Authors

Dear authors, 

greetings!

The manuscript was successfully improved. Attention was didicated to the highlighted aspects.

Reviewer 2 Report

Comments and Suggestions for Authors

#1 is exactly the validation experiment to confirm whether the tool is appropriately working.   The rest of responses are adequate.

Reviewer 3 Report

Comments and Suggestions for Authors

Dear authors, 

Thank you for taking your time and revise the manuscript according to my remarks and also following the other reviewers observations. 

I believe that the paper is significantly improved, you answered to all my remarks and from my point of view, after fine minor editing, the paper is suitable for publication. 

Comments on the Quality of English Language

Minor English editing - typos and small errors within the text. 
